# Causal Counterfactual Learning in Vision-Language for WSI Classification

## Abstract

Traditional Multiple Instance Learning (MIL), as a core method for Whole Slide Image (WSI) classification in computational pathology, often leads to model misclassification due to insufficient information when relying solely on visual representations. The introduction of Large Language Models (LLMs) has provided rich textual prompts to enhance visual representations. However, the data-driven learning of LLMs often induces spurious correlations between visual signals and text, causing inaccurate textual descriptions that pollute the alignment process and degrade WSI classification performance. To address this issue, we propose a Causal-learning Dual-attention MIL framework (CDMIL). The framework first achieves preliminary alignment through a prototype-guided dual-attention mechanism, followed by a counterfactual learning strategy for causal intervention. Replacing factual text with counterfactual text forces the model to abandon its reliance on spurious correlations and instead learn genuine causal relationships. Experiments demonstrate that CDMIL achieves state-of-the-art performance in both accuracy and out-of-distribution robustness, validating the superiority of this causal learning framework. The code will be released at https://github.com/xxx/CDMIL.

## 1 Introduction

The digital revolution in pathology has propelled computational pathology to the forefront of modern precision medicine (Wang et al., 2024; Tsai et al., 2023). The transformation of traditional glass slides into gigapixel-scale whole slide images (WSIs) has enabled digital pathology to create unprecedented opportunities for objective, reproducible, and high-throughput diagnostic analysis. In practice, however, assembling large-scale datasets with precise pixel-level annotations is often challenging and infeasible. To alleviate this limitation, Multiple Instance Learning (MIL) has been widely adopted as a mainstream learning paradigm (Lu et al., 2021; Zhang et al., 2022; Li et al., 2023a; Wang et al., 2022; Shao et al., 2021; Li et al., 2023b; Zhang et al., 2024; Shi et al., 2024).

MIL operates on small patches (also known as "instances") to construct a slide-level (or "bag-level") representation for analysis (Maron & Lozano-Pérez, 1997). An MIL-based approach typically follows a three-stage pipeline: (1) patches are cropped from the WSI, (2) a pre-trained encoder is used to extract patch features, and (3) these features are aggregated into a slide-level representation to perform WSI classification. Methods following this paradigm have achieved remarkable success in various pathological diagnostic tasks. However, a key limitation of conventional MIL is its exclusive reliance on visual signals, which often leads models to learn merely statistical correlations rather than the underlying pathological principles relied upon by human experts (Shi et al., 2024). The rapid emergence of large language models (LLMs) offers a promising avenue to address this shortcoming. Equipped with strong text-generation capabilities, LLMs can furnish rich, descriptive annotations for pathology images, thereby supplying the semantic context that purely visual pipelines lack (Shi et al., 2024). This progress has, in turn, catalyzed vision–language approaches in pathology that align visual features from WSI patches with corresponding diagnostic text prompts (Huang et al., 2023; Zheng et al., 2025; Shi et al., 2024). The central hypothesis is that guiding the model with explicit pathological concepts (e.g., "pleomorphic nuclei," "mitotic figures") enables it to learn representations that are more meaningful and clinically relevant. Nevertheless, while such vision–language alignment can enhance performance and interpretability, it introduces a critical challenge: ensuring that the alignment is robust not merely in a statistical sense but is firmly grounded in genuine pathological principles.

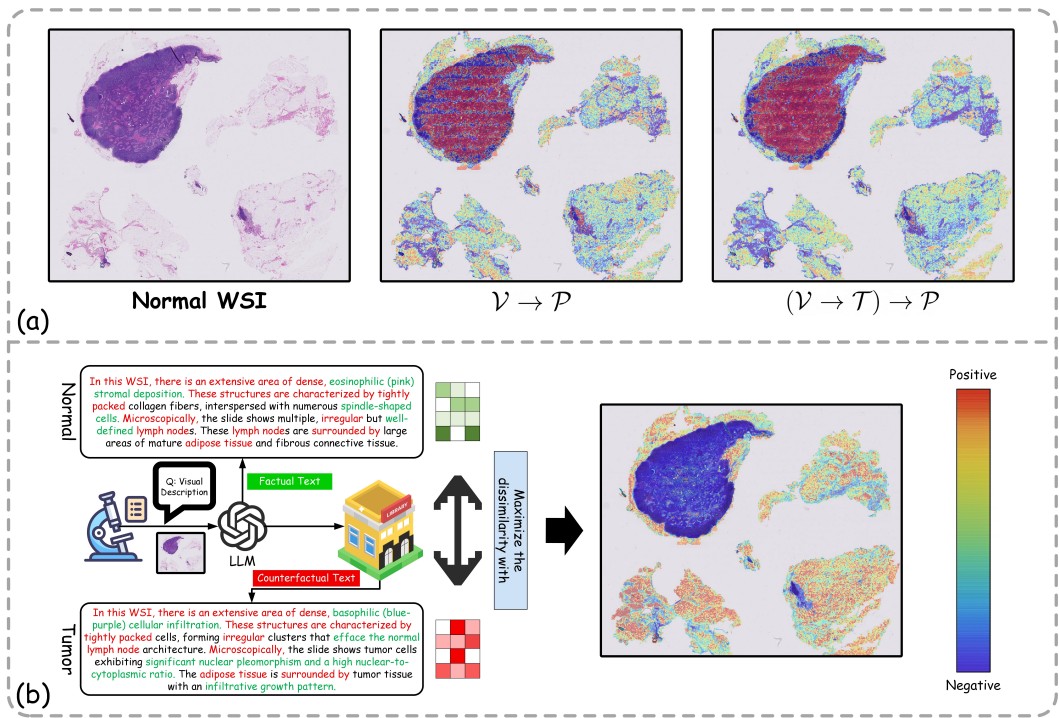

Figure 1: (a) This panel illustrates that when relying solely on WSI visual information ($\mathcal{V} \to \mathcal{P}$), the model may be influenced by spurious correlations in features, leading to misjudgment and focus on incorrect regions. When combined with textual descriptions ($(\mathcal{V} \to \mathcal{T}) \to \mathcal{P}$), erroneous textual associations can exacerbate this misjudgment, causing the model to further focus on irrelevant regions. (b) Through counterfactual causal learning, the model is able to avoid directing tumor-like attention to normal regions. The red text in the figure indicates that even though tumor and normal slides may have similar textual descriptions, whereas the green text represents the key textual descriptions that truly influence tumor identification.

Due to tumor-induced Extracellular Matrix remodeling and fibrotic reactions, these regions often exhibit visual similarities to tumorous areas in the training data (Yuan et al., 2023). Consequently, when an LLM "observes" the visual features of such a normal slide, it may generate a description that, in addition to normal tissue features, includes terms typically associated with tumor regions (e.g., "dense stromal changes," "desmoplasia"). This introduces spurious tumor-related information into the textual modality. A model trained to simply align with this "contaminated" text can be easily misled into misclassifying a normal slide as tumor, as illustrated in Fig. 1. This exemplifies a classic case of confounding, where the visual features of fibrotic stroma act as a confounder, creating a non-causal, spurious link between the "normal" slide and "tumor-like" text.

To address this challenge, we propose the Causal-learning Dual-attention Multiple Instance Learning (CDMIL) framework, coupling prototype-driven dual-attention vision–language alignment with a counterfactual causal learning strategy. Specifically, prototype-guided dual attention aligns visual features with text representations, while counterfactual constraints enforce prediction gaps pushing the model toward causal rather than spurious textual cues. In this way, CDMIL grounds its predictions in genuine pathological evidence over superficial data co-occurrence.

**Contributions.** To summarize, our contributions are:

- We design a prototype-driven dual-attention mechanism to achieve robust vision-language alignment with the descriptions generated by LLMs.

- We propose a counterfactual causal learning strategy aimed at enforcing vision-language causality while mitigating the model's reliance on spurious correlations in textual features.

- We demonstrate that our approach achieves SOTA performance on WSI classification tasks and exhibits exceptional generalization.

## 2 RELATED WORKS

**Multiple Instance Learning for WSI Classification.** MIL is a widely recognized approach to the classification of WSIs, especially in situations with weak supervision. In MIL, each WSI is represented as a "bag" $\mathcal{B} = \{x_1, x_2, \ldots, x_N\}$ of $N$ image patches at the instance level. Only a single slide-level label $Y \in \{0, \ldots, C-1\}$ is provided. The existing approaches in MIL can generally be divided into two broad categories: (1) explicit modeling, where the slide label is predicted by aggregating the predictions at the instance level, ie $\hat{Y} = \text{pool}\{\hat{y}_1, \hat{y}_2, \ldots, \hat{y}_N\}$, using pooling techniques such as mean or max pooling (Campanella et al., 2019; Zhang et al., 2022), and (2) implicit representation learning, where a slide level representation $z = f(\mathcal{B})$ is learned by combining embeddings of the instance, followed by classification using methods such as recurring neural networks (RNNs) (Campanella et al., 2019) or attention mechanisms (Ilse et al., 2018).

In the latter case, attention-based MIL methods are highly effective, learning adaptive instance weights to reduce inductive biases in complex histopathological patterns. Architectural innovations have further enhanced this, such as CLAM's clustering branch for feature separation (Lu et al., 2021) and TransMIL's self-attention for capturing inter-instance dependencies (Shao et al., 2021). Other strategies optimize the learning process by improving instance selection (Yu et al., 2023a; Li et al., 2023a), enhancing representations via contrastive sampling (Wang et al., 2022), or ensuring robustness against dominant patches (Zhang et al., 2024).

**Vision-Language Models for Computational Pathology.** The Vision Language Models (VLM) paradigm, exemplified by CLIP (Radford et al., 2021) and FLIP (Li et al., 2023b), has proven effective in various visual recognition tasks (Khattak et al., 2023; Yu et al., 2023b; Luo et al., 2025). These models use a dual-encoder architecture, aligning visual and textual inputs into a shared embedding space via pretraining on large datasets. In computational pathology, MI-Zero (Lu et al., 2023) and PLIP (Huang et al., 2023) have applied this paradigm to align visual features with pathology texts. However, curating large-scale, high-quality image-text datasets remains labor-intensive. LLMs offer a promising solution by generating rich textual descriptions for pathology images, potentially bypassing manual data collection and making large-scale dataset creation more accessible.

A fundamental challenge arises as LLMs generate descriptions based on statistical co-occurrence, not causal logic. This means they can encode spurious correlations, such as staining artifacts with disease labels, directly into the text (Lin et al., 2024a). Models trained to align with this text consequently inherit these biases, compromising their out-of-distribution (OOD) robustness (Lin et al., 2024b; Zhang & Ranganath, 2023). Our work introduces a counterfactual mechanism to directly address this. Replacing factual text with counterfactuals from a feature library challenges the model to move beyond simple alignment and validate the causal necessity of its learned vision-language associations, ensuring predictions are grounded in genuine pathological evidence.

## 3 PROBLEM SETUP & BACKGROUND

### 3.1 THE "CORRELATION" VS. "CAUSALITY" DILEMMA IN VISUAL RECOGNITION

Traditional supervised deep learning, especially in the field of computer vision, is essentially a powerful **Correlation Learning Engine**. Given a large-scale dataset

$$\mathcal{D} = \{(\mathcal{X}_i, \mathcal{Y}_i)\}_{i=1}^{N}, \tag{1}$$

the model learns parameters $\theta$ by minimizing an empirical risk function:

$$\min_{\theta} \frac{1}{N} \sum_{i=1}^{N} \mathcal{L}(f(\mathcal{X}_i; \theta), \mathcal{Y}_i), \tag{2}$$

where $f$ is the deep neural network, and $\mathcal{L}$ is the loss function. This process effectively learns the statistical conditional probability $P(\mathcal{Y}|\mathcal{X})$ from the data distribution $P(\mathcal{X}, \mathcal{Y})$. For example, the model learns to associate the visual pattern "furry ears and whiskers" with the label "cat".

However, this learning paradigm has a fundamental theoretical flaw: *correlation does not imply causation*. The strong correlations learned by the model may not reflect true causal relationships but are simply artifacts of confounding bias present in the dataset. Suppose there is an unobserved confounding variable $\mathcal{Z}$ that affects both the image features $\mathcal{X}$ and the labels $\mathcal{Y}$. In this case, the model learns $P(\mathcal{Y}|\mathcal{X})$ as a weighted average across various confounded environments:

$$P(\mathcal{Y}|\mathcal{X}) = \sum_z P(\mathcal{Y}|\mathcal{X}, \mathcal{Z} = z)P(\mathcal{Z} = z|\mathcal{X}). \tag{3}$$

A classic example is the "cow on the grass" problem (Beery et al., 2018): if the confounding variable $\mathcal{Z}$ represents "outdoor grass", the model might mistakenly treat the "green background" as a feature that is falsely correlated rather than a causal feature for recognizing "cow". When the model is tested on a new distribution $P'(\mathcal{Z}|\mathcal{X})$ (e.g., beach, indoor), its performance will drop dramatically due to the reliance on spurious correlations (Fan et al., 2025). This phenomenon reveals the vulnerability of existing models in OOD generalization. Moreover, LLMs may generate text that falsely correlates with visual signals, further reinforcing the model's reliance on non-causal associations.

## 3.2 Causal Inference Theory: Structural Causal Models and Interventions

To formalize *causality* mathematically, we introduce the theoretical framework of Structural Causal Models (SCM) (Pearl, 2010). An SCM describes the world as a graph (Directed Acyclic Graph, DAG) consisting of variable nodes and directed edges, where the edges represent direct causal relationships between variables. In our vision-language task, a simplified causal graph can be represented as: visual features $\mathcal{V}$ lead to text features $\mathcal{T}$, and both jointly influence the final prediction $\mathcal{P}$, as shown in Fig. 2.

Traditional supervised learning, i.e., learning $(\mathcal{P}|\mathcal{V},\mathcal{T})$ in Eq. 1, involves observational prediction. However, true causal problems concern interventional prediction, i.e., $(\mathcal{P}|do(\mathcal{T} = t))$, where the *do*-operator (Pearl, 2010) represents a forcible "intervention": we set $\mathcal{T} = t$ and block all causal paths leading to $\mathcal{T}$ (for example, replacing the original text input in our model), then observe how $\mathcal{P}$ changes.

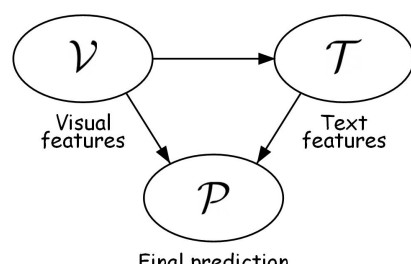

In real-world visual tasks, we cannot arbitrarily perform *do*-interventions on the input data as in physical experiments. Therefore, researchers have proposed a method called *causal regularization*, where the core idea is to introduce specific constraints in the model's learning process to approximate or encourage the model to behave like a causal model undergoing intervention (Schölkopf et al., 2021; Arjovsky et al., 2019).

Figure 2: Simplified causal graph showing the relationship between visual features $\mathcal{V}$, text features $\mathcal{T}$, and the final prediction $\mathcal{Y}$ in a vision-language task.

*Counterfactual reasoning* is one of the most powerful tools for implementing causal regularization. It explores the question: "What if things had been different?" This can be viewed as a "thought experiment" conducted within the model, serving as a mathematical approximation of the *do*-intervention.

In our work, this idea is concretized as a counterfactual causal loss ($\mathcal{L}_{\text{causal}}$), which is combined with the standard supervised loss. The specific implementation details are presented in Sec. 4.2 and 4.3.

## 4 Methodology

In this section, we detail our proposed causal learning framework, summarized in Fig. 3. Sec. 4.1 introduces the core CDMIL architecture, which aligns visual features with text descriptions from LLMs via a prototype-driven dual-attention mechanism, before using mean pooling for prediction. Sec. 4.2 describes our causal intervention strategy, where a text feature library is used to generate counterfactual samples by replacing the factual text prompt. Sec. 4.3 presents the causal regularization loss, which maximizes the divergence between factual and counterfactual predictions to penalize reliance on spurious correlations, thereby enforcing a robust, causal vision-language alignment.

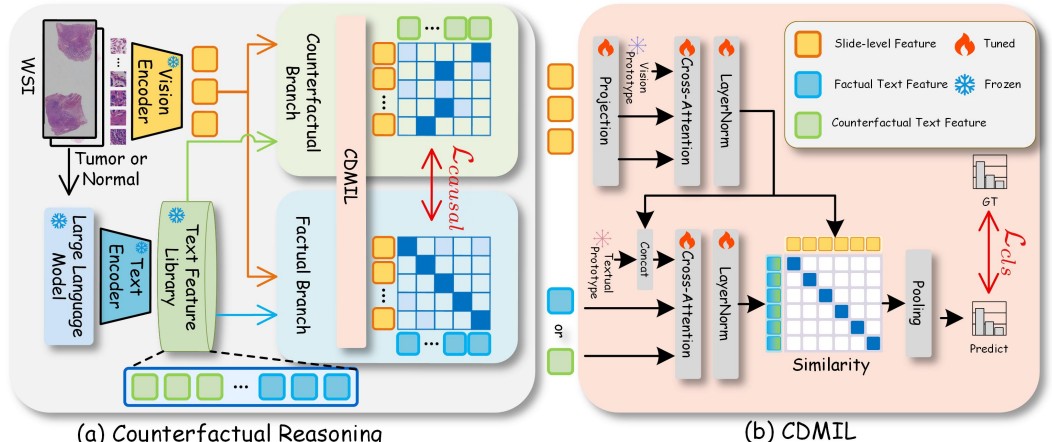

(a) Counterfactual Reasoning

(b) CDMIL

Figure 3: The architecture of our proposed CDMIL. (a) The training scheme features two parallel streams: a Factual Branch with the original prompt, and a Counterfactual Branch where the prompt is replaced by a prototype from the Text Feature Library. The causal loss $\mathcal{L}_{\text{causal}}$ enforces inconsistency between the two branches' predictions. (b) The main pipeline uses a dual cross-attention mechanism to fuse visual features and textual features via learnable prototypes. The prediction is obtained through pooling, and the classification loss $\mathcal{L}_{\text{cls}}$ is computed with respect to the ground truth (GT).

## 4.1 DUAL-ATTENTION VISION-LANGUAGE ALIGNMENT

Given a WSI, we first represent it as a bag of instance features $\mathcal{V} = \{v_1, v_2, \ldots, v_N\} \in \mathbb{R}^{N \times D_v}$, extracted by a pre-trained ResNet-50 (He et al., 2015). Simultaneously, a pre-trained text encoder (Radford et al., 2021) processes the text descriptions obtained through LLMs to extract text features $\mathcal{T} \in \mathbb{R}^{B \times D_t}$, where $B$ denotes the number of text prompts. The core of our model is a two-stage cross-attention mechanism designed to synergize these modalities.

**Stage 1: Visually-Grounded Prototype Refinement.** We initialize a set of learnable, class-specific visual prototypes $P_{\text{vis}} \in \mathbb{R}^{C \times D_v}$, where $C$ is the number of classes. These prototypes act as high-level conceptual queries. We use a cross-attention layer to allow these prototypes to "attend to" the bag of instance features, aggregating the most relevant visual information for each class concept:

$$P'_{\text{vis}, \_} = \text{CrossAttention}_1 \left( \text{query} = P_{\text{vis}}, \text{key} = \mathcal{V}, \text{value} = \mathcal{V} \right) \quad (4)$$

where $P'_{\text{vis}}$ are the refined, context-aware visual prototypes. A residual connection and layer normalization are applied to obtain the final visual slide features.

**Stage 2: Text-Modulated Fusion.** We then fuse the refined slide features with the text features $\mathcal{T}_{\text{fact}}$. To do this, we first concatenate the slide features with another set of learnable textual prototypes $P_{\text{text}} \in \mathbb{R}^{C \times D_t}$ (where $D_t = D_v$) to form a combined query matrix $Q_{\text{text}} \in \mathbb{R}^{2C \times D_t}$. A second cross-attention layer then uses this query to interact with the text features:

$$\mathcal{T}_{\text{fused}} = \text{CrossAttention}_2 \left( \text{query} = Q_{\text{text}}, \text{key} = \mathcal{T}, \text{value} = \mathcal{T} \right) \quad (5)$$

The final fused representation is obtained by another residual connection and normalization. This representation is then used to compute the similarity matrix $S$ with slide features, and the final classification logits are obtained via Mean Pooling.

## 4.2 COUNTERFACTUAL INTERVENTION FOR CAUSAL LEARNING

The fusion mechanism described above excels at learning correlations, but is susceptible to spurious ones. To address this, we introduce a counterfactual learning scheme to enforce causal reasoning. This process is active only during training and after an initial "burn-in" epoch.

**Step 1: Text Feature Library Population (Epoch 1).** During the first training epoch, we only perform standard supervised learning. Concurrently, for each training sample $(\mathcal{V}, \mathcal{T}, \mathcal{Y})$, we compute its text features $\mathcal{T}_{\text{fact}}$ and accumulate them in a temporary library, grouped by the ground-truth label $\mathcal{Y}$. At the end of the epoch, we compute the mean feature vector for each class to populate our main online text feature library,

$$L_{\text{text}} = \{\bar{\mathcal{T}}_0, \bar{\mathcal{T}}_1, \ldots, \bar{\mathcal{T}}_{C-1}\}. \tag{6}$$

**Step 2: Counterfactual Intervention (Epoch $> 1$).** For subsequent epochs, for each training sample, we perform an intervention on the text modality. While keeping the visual features $\mathcal{V}$ unchanged, we replace the factual text features $\mathcal{T}_{\text{fact}}$ with a counterfactual text feature $\mathcal{T}_{\text{cf}}$. This $\mathcal{T}_{\text{cf}}$ is retrieved from our library $L_{\text{text}}$ corresponding to a different class:

$$\mathcal{T}_{\text{cf}} = L_{\text{text}}[(\mathcal{Y} + 1) \mod C]. \tag{7}$$

**Step 3: Computing Counterfactual Prediction.** We then pass the contradictory pair $(\mathcal{V}, \mathcal{T}_{\text{cf}})$ through the exact same dual-attention fusion pipeline as the factual branch to obtain the counterfactual logits, $\text{logits}_{\text{cf}}$. This ensures that the visual signal remains unchanged while only the textual input varies, thereby blocking the causal path from $\mathcal{V}$ to $\mathcal{T}$ and enabling any change in prediction to be attributed solely to the textual intervention.

### 4.3 Joint Optimization with Causal Loss

Our total loss function $\mathcal{L}_{\text{total}}$ is defined as:

$$\mathcal{L}_{\text{total}} = \mathcal{L}_{\text{cls}} + \lambda \cdot \mathcal{L}_{\text{causal}}, \tag{8}$$

where $\lambda$ is a balancing hyperparameter ($\lambda = 0.1$).

*Supervised loss $\mathcal{L}_{\text{cls}}$:* This is the standard classification loss applied to the observational data. Let $f$ be our model, and let $\text{logits}_{\text{fact}} = f(\mathcal{V}, \mathcal{T}_{\text{fact}})$ be the output of the factual branch, then:

$$\mathcal{L}_{\text{cls}} = \text{CrossEntropy}(\text{logits}_{\text{fact}}, \mathcal{Y}_{\text{true}}) \tag{9}$$

**Causal Regularization on Predictions:** The causal loss $\mathcal{L}_{\text{causal}}$ is designed to enforce causal reasoning by ensuring that the model's final prediction is sensitive to interventions on its inputs. This term aims to maximize the divergence between the factual and counterfactual prediction distribution.

**Computing Factual and Counterfactual Predictions:** We pass both the factual pair $(\mathcal{V}, \mathcal{T}_{\text{fact}})$ and counterfactual pair $(\mathcal{V}, \mathcal{T}_{\text{cf}})$ through the entire model $f_\theta$ to obtain their respective final logits:

$$\text{logits}_{\text{fact}} = f_\theta(\mathcal{V}, \mathcal{T}_{\text{fact}}) \quad \text{and} \quad \text{logits}_{\text{cf}} = f_\theta(\mathcal{V}, \mathcal{T}_{\text{cf}}) \tag{10}$$

The intervention is isolated to the textual modality, as the visual evidence $\mathcal{V}$ is held constant.

**Defining the Causal Loss:** A causally-aware model, when presented with a contradictory pair like (cancer-image, normal-text-prototype), should produce a significantly different final prediction compared to the factual case. We enforce this by maximizing the distance between the two prediction distributions. We first convert the logits into probability distributions using the softmax function, and then use the negative KL-divergence as our loss:

$$\mathcal{L}_{\text{causal}} = -D_{\text{KL}}(\text{softmax}(\text{logits}_{\text{cf}}) \parallel \text{softmax}(\text{logits}_{\text{fact}})) \tag{11}$$

The minimization of the total loss in Eq. 8 serves a dual purpose: it ensures the model fits the observational data (via $\mathcal{L}_{\text{cls}}$) while, more critically, imposing causal regularization on its predictive behavior. This regularization penalizes "lazy" models that are insensitive to semantic interventions on their inputs. It encourages the model to ground its predictions in true, robust causal evidence rather than relying on spurious correlations arising from data bias. Ultimately, this enables the model to achieve superior out-of-distribution generalization, which is the core value of causal learning.

## 5 Experiments

### 5.1 Experimental setup

**Datasets.** To comprehensively evaluate the efficacy of our proposed CDMIL framework and its counterfactual causal learning mechanism, we designed an experimental protocol using a suite of

histopathology datasets. For the prediction of Epidermal Growth Factor Receptor (EGFR) mutations in Lung Adenocarcinoma (LUAD), we utilized two in-house datasets from two independent hospitals, $LCEM_1$ ($n$=777) and $LCEM_2$ ($n$=844), supplemented by the public TCGA-LUAD dataset ($n$=528). Additionally, the Camelyon16 dataset ($n$=399) was employed for the binary classification of breast cancer lymph node metastases (tumor vs. normal). In our experimental design, the $LCEM_1$ and Camelyon16 datasets were used to train models and compare their performance against baseline and state-of-the-art methods. To rigorously assess zero-shot generalization, models trained on $LCEM_1$ were directly evaluated on the unseen $LCEM_2$ and TCGA-LUAD datasets.

**Implementation Details.** For preprocessing, we extracted non-overlapping $256 \times 256$ patches at $20\times$ magnification from tissue regions identified by Otsu's thresholding. All experiments ran on an NVIDIA RTX 4090 GPU, using a ResNet-50 (He et al., 2015) pre-trained on ImageNet as the feature encoder. The model was optimized with Adam (weight decay=1e-5), using an initial learning rate of 3e-4 (later reduced to 1e-4) and an early stopping patience of 20. For $LCEM_1$, we used an 80:20 random split for training/testing, with the training set further split 8:2 for validation. For the Camelyon16 dataset, however, we strictly adhered to the official protocol and used its predefined test set for all final evaluations.

**Evaluation Metrics.** We evaluated all methods using three key metrics: the Area Under the Receiver Operating Accuracy (ACC), Characteristic curve (AUC), and F1-score. To ensure robust evaluation, each method was tested five times, with the dataset being randomly partitioned for each run according to predefined ratios. The final results are reported as the mean and standard deviation of these five runs. Furthermore, a paired t-test was used to determine if the performance difference between two methods was statistically significant. A p-value greater than 0.05 was considered to indicate that the difference between the results was not statistically significant.

## 5.2 Performance comparison with existing works

As presented in Table 1, our proposed CDMIL method achieves state-of-the-art (SOTA) performance on both datasets, attaining an AUC of 88.11% on $LCEM_1$ and 90.84% on Camelyon16. These results demonstrate a consistent and significant advantage over all competing methods. The superiority of our approach is particularly pronounced on the more challenging $LCEM_1$ dataset. We attribute this success to the integration of textual information with counterfactual causal learning, which enables the model to focus on more discriminative features and thereby enhances its classification capability.

Table 1: Comparison of WSI classification performance (mean % $\pm$ std). **Bold** indicates the best result, an underline indicates the second best, and * denotes comparable performance to the top result (paired t-test, p > 0.05).

| Method | $LCEM_1$ | | | Camelyon16 | | |
|---|---|---|---|---|---|---|
| | ACC | AUC | F1 | ACC | AUC | F1 |
| MeanPooling (Campanella et al., 2019) | $62.21_{\pm6.82}$ | $62.11_{\pm7.00}$ | $42.60_{\pm11.69}$ | $74.23_{\pm8.12}$ | $75.56_{\pm9.33}$ | $72.14_{\pm8.91}$ |
| MaxPooling (Campanella et al., 2019) | $64.36_{\pm7.42}$ | $66.68_{\pm9.78}$ | $46.89_{\pm14.73}$ | $73.78_{\pm9.66}$ | $75.34_{\pm9.88}$ | $72.07_{\pm7.67}$ |
| CLAM (Lu et al., 2021) | $68.74_{\pm7.23}$ | $70.27_{\pm11.68}$ | $60.09_{\pm15.06}$ | $80.78_{\pm4.73}$ | $83.04_{\pm4.48}$ | $78.29_{\pm5.63}$ |
| ABMIL (Ilse et al., 2018) | $66.75_{\pm9.75}$ | $69.00_{\pm8.22}$ | $54.80_{\pm17.19}$ | $76.74_{\pm9.07}$ | $78.36_{\pm7.96}$ | $74.37_{\pm18.00}$ |
| DSMIL (Li et al., 2021) | $68.28_{\pm3.76}$ | $69.75_{\pm4.20}$ | $63.37_{\pm4.36}$ | $75.35_{\pm6.31}$ | $78.26_{\pm6.75}$ | $73.41_{\pm8.95}$ |
| TransMIL (Shao et al., 2021) | $60.40_{\pm4.82}$ | $54.85_{\pm5.15}$ | $37.61_{\pm1.93}$ | $78.33_{\pm7.23}$ | $79.25_{\pm8.17}$ | $77.92_{\pm9.87}$ |
| PMIL (Yan et al., 2025) | $\underline{73.16}_{\pm1.79}$* | $\underline{82.29}_{\pm1.84}$* | $\underline{72.27}_{\pm1.87}$* | $\underline{83.88}_{\pm0.90}$* | $\underline{87.27}_{\pm2.36}$* | $\underline{81.84}_{\pm1.60}$* |
| CDMIL | $\mathbf{78.58}_{\pm1.18}$ | $\mathbf{88.11}_{\pm1.67}$ | $\mathbf{75.63}_{\pm1.27}$ | $\mathbf{87.66}_{\pm1.10}$ | $\mathbf{90.84}_{\pm3.09}$ | $\mathbf{85.15}_{\pm5.80}$ |

## 5.3 Zero-Shot Generalization Performance

To rigorously evaluate the model's generalization capability, we deployed models trained on the $LCEM_1$ dataset directly onto two external test sets: $LCEM_2$ and TCGA-LUAD. As detailed in Table 2, the results unequivocally demonstrate the exceptional OOD performance of CDMIL. On the $LCEM_2$ dataset, CDMIL achieves an AUC of 85.95%, surpassing the next-best baseline, ABMIL (62.47%), by a substantial margin of over 23 percentage points. This superiority is further confirmed on the highly challenging TCGA-LUAD dataset, which is characterized by severe class imbalance; here, CDMIL attains an AUC of 86.32% while all other methods remain below 64%, showcasing its

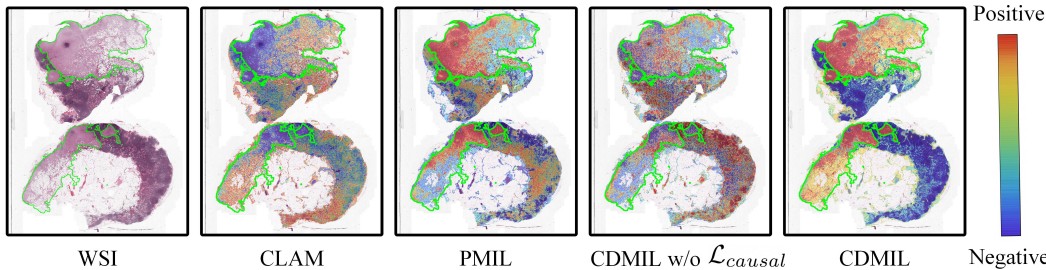

WSI       CLAM       PMIL       CDMIL w/o $\mathcal{L}_{causal}$       CDMIL

Figure 4: Visualization of instance-level attention on slide *test_090* from the Camelyon16 dataset. (green contour: GT)

remarkable robustness. Crucially, when the counterfactual causal learning mechanism is removed, the model's performance degrades dramatically. This provides compelling evidence that the introduction of text prompts alone is insufficient to ensure robust generalization. Without the constraint of causal regularization, even a vision-language model is susceptible to learning spurious correlations. In summary, the consistent and significant performance gap demonstrates that both traditional MIL approaches and simple vision-text alignment models are prone to overfitting spurious correlations in the training data, which fail to generalize to new distributions.

Table 2: **Zero-shot** generalization performance comparison on the EGFR mutation prediction task. CDMIL w/o $\mathcal{L}_{causal}$ represents an ablation variant of our model without the causal learning mechanism. (**Notably, due to a severe class imbalance of EGFR mutation types in the TCGA-LUAD dataset, AUC and F1 is considered the primary metrics for a fair comparison on this dataset.**)

| Method | LCEM$_2$ | | | TCGA-LUAD | | |
|---|---|---|---|---|---|---|
| | ACC | AUC | F1 | ACC | AUC | F1 |
| MeanPooling | 61.73 | 59.85 | 38.17 | 14.02 | 58.71 | 12.29 |
| MaxPooling | 59.12 | 55.66 | 55.16 | 56.25 | 58.25 | 48.84 |
| CLAM | 42.54 | 58.55 | 38.22 | **84.66** | 63.58 | 51.30 |
| ABMIL | 61.73 | 62.47 | 38.17 | 14.02 | 59.50 | 12.29 |
| DSMIL | 50.36 | 60.78 | 50.13 | 80.30 | 61.51 | 54.38 |
| TransMIL | 66.55 | 61.73 | 48.17 | 14.02 | 46.97 | 12.29 |
| PMIL | 54.38 | 61.78 | 52.36 | 80.68 | 60.91 | 50.04 |
| CDMIL w/o $\mathcal{L}_{causal}$ | 63.51 | 58.26 | 51.65 | 57.77 | 60.29 | 49.10 |
| CDMIL | **70.73** | **85.95** | **65.39** | 81.06 | **86.32** | **56.67** |

## 5.4 QUALITATIVE RESULTS AND VISUALIZATION

As illustrated in Fig. 4, the positive predictions generated by CDMIL demonstrate a strong correspondence with the ground-truth annotations, which is reflected in a higher intersection over union and greater prediction confidence scores. In contrast, the CDMIL variant without counterfactual causal learning exhibits erroneous predictions, misallocating attention to non-tumorous regions. Furthermore, we visualize the slide-level feature clusters on the LCEM$_1$ and Camelyon16 test sets using t-SNE (Van der Maaten & Hinton, 2008) in Fig. 5. Compared to the best-performing baseline MIL methods, the embedding space learned by CDMIL exhibits superior intra-class compactness and inter-class separability. This further substantiates the superiority of our approach by visually demonstrating that it learns more discriminative representations.

## 5.5 ABLATION STUDY

To dissect the contributions of each key component, we conducted a comprehensive ablation study (see Table 3). A noteworthy phenomenon is that the model incorporating only textual information

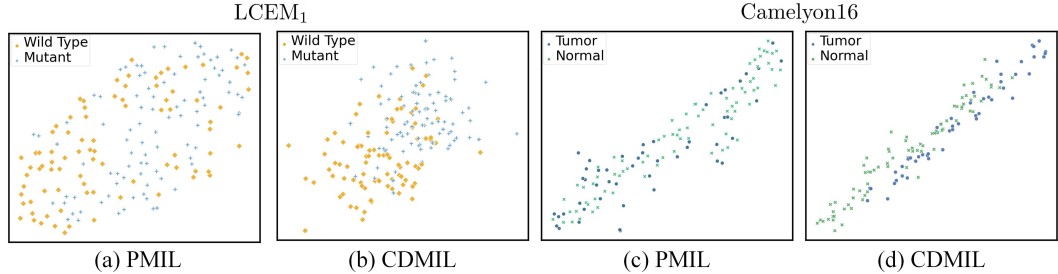

$LCEM_1$         Camelyon16

(a) PMIL       (b) CDMIL       (c) PMIL       (d) CDMIL

Figure 5: Visualization of slide-level feature clustering results using t-SNE with different methods on the $LCEM_1$ and Camelyon16 test sets.

exhibits high variance, which indicates that while simple vision-text alignment can raise the performance ceiling, it also makes the model unstable and prone to overfitting spurious correlations present in the text across different data splits. However, after introducing the counterfactual causal learning mechanism, the model's performance not only improves substantially, but its variance is also significantly reduced. This provides strong evidence that our causal regularization, acting as a constraint, effectively stabilizes the training process, forcing the model to ignore spurious associations and focus on learning more robust causal features. Therefore, while high-quality text (with GPT-5 performing best) serves as the foundation, the causal learning mechanism is the key to achieving both high mean performance and high stability.

Table 3: Ablation study of CDMIL's key components on three datasets. We evaluate the incremental benefits of integrating Text, our Causal Learning mechanism, and different LLMs over a Vision-only baseline. Results are reported as classification AUC (mean % ± std).

| Vision | Text | $\mathcal{L}_{casual}$ | Gemini-2.5 | GPT-4.1 | GPT-5 | $LCEM_1$ | Camelyon16 |
|:---:|:---:|:---:|:---:|:---:|:---:|:---:|:---:|
| ✓ | | | | | | $69.09_{\pm 4.38}$ | $76.87_{\pm 5.81}$ |
| ✓ | ✓ | | ✓ | | | $75.81_{\pm 10.69}$ | $79.24_{\pm 8.96}$ |
| ✓ | ✓ | | | ✓ | | $81.70_{\pm 11.33}$ | $80.13_{\pm 9.81}$ |
| ✓ | ✓ | | | | ✓ | $84.23_{\pm 12.50}$ | $82.79_{\pm 10.86}$ |
| ✓ | ✓ | ✓ | ✓ | | | $77.32_{\pm 5.63}$ | $81.54_{\pm 6.28}$ |
| ✓ | ✓ | ✓ | | ✓ | | $86.51_{\pm 3.15}$ | $84.37_{\pm 2.33}$ |
| ✓ | ✓ | ✓ | | | ✓ | $\mathbf{88.11_{\pm 1.67}}$ | $\mathbf{90.84_{\pm 3.09}}$ |

## 6    CONCLUSION AND DISCUSSION

In this paper, we addressed the challenge of spurious correlations in vision-language computational pathology, which undermines OOD robustness. We proposed CDMIL, a framework that integrates a counterfactual learning mechanism into a vision-language alignment model. CDMIL challenges the model by dynamically replacing factual text with counterfactual features from a text feature library, regularizing it to ground predictions in causally robust evidence. Experiments on pathology classification tasks showed that CDMIL achieves state-of-the-art accuracy and, critically, demonstrates significantly improved OOD robustness over existing methods. These results demonstrate that enforcing causal reasoning through counterfactual intervention is an effective strategy, and the framework for integrating causal reasoning into multimodal MIL contributes to the development of more reliable and interpretable AI systems in computational pathology.

**Limitation and Future works.** This study has several limitations that suggest directions for future research. Our current counterfactual interventions are applied only to the textual modality. A natural extension would be to develop methods for creating visual counterfactuals (Liu et al., 2025). Additionally, the feature library could be improved by incorporating more sophisticated update mechanisms (He et al., 2020).

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

# APPENDIX

## A    THEORETICAL BACKGROUND ON COUNTERFACTUAL CAUSAL INFERENCE

o provide a deeper theoretical foundation for the causal learning mechanism employed in our CDMIL framework, we briefly review the principles of counterfactual causal inference, primarily based on the structural causal model (SCM) framework developed by Judea Pearl (Pearl, 2010).

### A.1    THE THREE LAYERS OF THE CAUSAL HIERARCHY

Causal inference theory posits a three-layer hierarchy of reasoning, each level enabling answers to a different class of questions:

**Layer 1: Association (Seeing)** This is the level of standard statistical and machine learning. It deals with purely observational data and answers questions about correlation, such as "What is the probability of a diagnosis $\mathcal{Y}$ given we observe visual features $\mathcal{V}$?" denoted as $P(\mathcal{Y}|\mathcal{V})$. Most deep learning models operate exclusively at this layer.

**Layer 2: Intervention (Doing)** This layer deals with questions about the effects of actively changing or "doing" something in the system. For example, "What would the diagnosis $\mathcal{Y}$ be if we forced the text description $T$ to be 'normal'?" denoted as $\mathcal{P}(\mathcal{Y}|do(\mathcal{T} = \text{'normal'}))$. The $do$-operator signifies a "hard" intervention that removes any pre-existing causal influences on the variable $\mathcal{T}$. Answering such questions is crucial for understanding causal effects, as it allows us to distinguish causation from spurious correlation.

**Layer 3: Counterfactuals (Imagining)** This is the highest level of causal reasoning. It deals with retrospective, "what if" questions that involve imagining a world that contradicts what was actually observed. For example, "Given that a patient with visual features $\mathcal{V}$ and factual text $\mathcal{T}_{\text{fact}}$ was diagnosed with cancer ($\mathcal{Y} = 1$), what would the diagnosis have been if the text had been 'normal' ($\mathcal{T} = \text{'normal'}$) instead?" This is denoted as $\mathcal{P}(\mathcal{Y}_{\mathcal{T}=\text{'normal'}}|\mathcal{V}, \mathcal{T}_{\text{fact}}, \mathcal{Y} = 1)$. Counterfactuals are more powerful than interventions because they use evidence from the specific, observed world (e.g., the specific visual features $\mathcal{V}$) to refine the prediction in a hypothetical world.

### A.2    COUNTERFACTUALS AS A TOOL FOR CAUSAL REGULARIZATION

In complex domains like vision and language, performing a true $do$-intervention on the data is often infeasible. We cannot, for instance, perfectly change a "cancerous" region in an image to a "normal" one without altering other correlated features.

Our work leverages Layer 3 reasoning—counterfactuals—as a powerful computational tool to approximate the goal of Layer 2. The counterfactual learning mechanism in CDMIL can be understood

as a form of causal regularization. Instead of directly manipulating the data, we simulate an intervention within the model's forward pass.

The process unfolds as follows:

- **Observation:** The model first processes the factual pair $(\mathcal{V}, \mathcal{T}_{\text{fact}})$ through its factual branch. This can be seen as the model establishing its "belief" about the world as it is, based on the observed evidence. The resulting prediction, $\text{logits}_{\text{fact}}$, is a Layer 1 (associational) outcome.

- **Imagining a Counterfactual World:** We then construct a counterfactual scenario. For the same visual evidence $\mathcal{V}$, we ask the model to imagine a world where the textual evidence was different. We achieve this by replacing $\mathcal{T}_{\text{fact}}$ with a counterfactual prototype $\mathcal{T}_{\text{cf}}$. This step is a direct implementation of a counterfactual query.

- **Causal Constraint:** The core of our approach lies in the causal loss, $\mathcal{L}_{\text{causal}} = -D_{\text{KL}}(\mathcal{P}_{\text{cf}} \parallel \mathcal{P}_{\text{fact}})$. This loss function does not simply measure a difference; it imposes a structural constraint on the model's reasoning process. It essentially tells the model: "Your prediction in the counterfactual world $\mathcal{P}_{\text{cf}}$ must be logically consistent with the change I made. If your initial prediction $\mathcal{P}_{\text{fact}}$ was heavily reliant on a spurious correlation between some visual artifact in $\mathcal{V}$ and the concept in $\mathcal{T}_{\text{fact}}$, then your prediction should not change much when I replace $\mathcal{T}_{\text{fact}}$ with $\mathcal{T}_{\text{cf}}$. I will penalize you for this 'stubbornness' by making the loss high."

Minimizing this loss extends beyond merely training the model for accuracy on observed data. We are regularizing its internal representations, forcing it to learn a function that is sensitive to causal interventions. This encourages the model to ground its predictions in features that are robustly linked across both the factual and counterfactual worlds—that is, the true causal features. This approach, therefore, helps the model to climb from Layer 1 (Association) towards a more robust, Layer 2-like (Interventional) predictive capability, leading to improved out-of-distribution generalization.

## B    DETAILED TRAINING ALGORITHM

To provide a comprehensive and reproducible overview of our training procedure, we present the detailed step-by-step algorithm for our Online Counterfactual Learning framework in Algorithm 1. This algorithm elaborates on the methodology described in the main paper, particularly detailing the two-stage training process. The key components of the algorithm are as follows:

**Epoch 1 (Lines 4-10):** The first epoch is dedicated to standard supervised learning and the initial population of the text feature library. The model is trained solely on the classification loss ($\mathcal{L}_{\text{cls}}$) while simultaneously accumulating the text features corresponding to each ground-truth class. At the end of this epoch, the main text feature library ($L_{\text{text}}$) is finalized by averaging the accumulated features for each class.

**Epochs $> 1$ (Lines 11-23):** From the second epoch onwards, the full causal learning mechanism is activated. For each batch, the model computes both a factual prediction and a counterfactual prediction. The counterfactual prediction is generated by intervening on the textual input, replacing it with a feature retrieved from the previously constructed library ($L_{\text{text}}$).

**Joint Optimization (Line 22):** The model is then optimized using a combined loss function, which includes both the standard classification loss and our proposed causal loss ($\mathcal{L}_{\text{causal}}$). This causal loss penalizes the model if its prediction is insensitive to the counterfactual intervention, thereby encouraging the learning of robust, causal associations.

This two-stage approach ensures that a stable and representative text feature library is constructed before the counterfactual regularization is applied, contributing to the overall stability and effectiveness of the training process.

---

**Algorithm 1** Counterfactual Learning for CDMIL

---

**Input:** Training data $\mathcal{D}$, model $f_\theta$, epochs $E$, causal weight $\lambda$.
**Output:** Causally-aligned model parameters $\theta$.
 1: Initialize model parameters $\theta$ and an empty text feature library $L_{\text{text}}$.
 2: **for** epoch $e = 1, \ldots, E$ **do**
 3:    **if** $e = 1$ **then**
 4:      **// Epoch 1: Supervised Learning & Library Population**
 5:      **for** each batch $(\mathcal{V}, \mathcal{T}, \mathcal{Y})$ in $\mathcal{D}$ **do**
 6:        Compute factual logits $\text{logits}_{\text{fact}} \leftarrow f_\theta(\mathcal{V}, \mathcal{T})$.
 7:        Compute classification loss $\mathcal{L}_{\text{cls}} \leftarrow \text{Loss}(\text{logits}_{\text{fact}}, \mathcal{Y})$.
 8:        Update parameters $\theta$ using $\mathcal{L}_{\text{cls}}$.
 9:        Accumulate text features derived from $\mathcal{T}$ into a temporary library based on $\mathcal{Y}$.
10:      **end for**
11:      Populate the main library $L_{\text{text}}$ by averaging accumulated features.
12:    **else**
13:      **// Epoch > 1: Counterfactual Causal Alignment**
14:      **for** each batch $(\mathcal{V}, \mathcal{T}, \mathcal{Y})$ in $\mathcal{D}$ **do**
15:        *// Factual Path*
16:        $\text{logits}_{\text{fact}} \leftarrow f_\theta(\mathcal{V}, \mathcal{T})$.
17:        $\mathcal{L}_{\text{cls}} \leftarrow \text{Loss}(\text{logits}_{\text{fact}}, \mathcal{Y})$.
18:        *// Counterfactual Path*
19:        Determine counterfactual label $\mathcal{Y}_{\text{cf}}$.
20:        Retrieve counterfactual text feature $\mathcal{T}_{\text{cf}} \leftarrow L_{\text{text}}[\mathcal{Y}_{\text{cf}}]$.
21:        $\text{logits}_{\text{cf}} \leftarrow f_\theta(\mathcal{V}, \mathcal{T}_{\text{cf}})$ {Visual input $\mathcal{V}$ remains unchanged}
22:        *// Causal Loss on Final Predictions*
23:        $\mathcal{L}_{\text{causal}} \leftarrow -D_{KL}(\text{softmax}(\text{logits}_{\text{cf}})||\text{softmax}(\text{logits}_{\text{fact}}))$. {Constraint on prediction space}
24:        *// Combined Update*
25:        $\mathcal{L}_{\text{total}} \leftarrow \mathcal{L}_{\text{cls}} + \lambda \cdot \mathcal{L}_{\text{causal}}$.
26:        Update parameters $\theta$ using $\mathcal{L}_{\text{total}}$.
27:      **end for**
28:    **end if**
29: **end for**

---

## C EXTERNAL EXPERIMENTAL RESULTS

### C.1 ADDITIONAL DATASET

To further evaluate the generalizability and robustness of our CDMIL framework across diverse histopathological tasks, we additionally incorporated the Breast Cancer Subtyping (BRACS) dataset. BRACS is a public, high-resolution WSI benchmark comprising 547 annotated images from 189 unique patients. It is designed for fine-grained, multi-class classification, aiming to distinguish between three core pathological categories: Benign, Atypical, and Malignant. This provides an ideal platform for validating model performance in complex, beyond-binary scenarios. The preprocessing for this dataset was conducted in accordance with the "Implementation Details" described in Sec. 5.1 of the main paper.

### C.2 PERFORMANCE ON LCEM$_2$ AND BRACS DATASETS

To provide a more comprehensive evaluation of our proposed CDMIL framework, we conducted additional comparative experiments on the LCEM$_2$ and BRACS datasets. The LCEM$_2$ dataset was utilized to further assess performance on the LUAD EGFR mutation prediction task, while the multi-class BRACS dataset was used to test the model's applicability to more complex classification scenarios. All experimental setups, including data preprocessing and evaluation metrics, were kept consistent with those described in the main paper.

The detailed quantitative results are presented in Table 4. On both datasets, our CDMIL method consistently outperforms all baseline and state-of-the-art MIL approaches across nearly all met-

Table 4: Comparison of WSI classification performance on $LCEM_2$ and BRACS (mean % $\pm$ std). **Bold** indicates the best result, an underline indicates the second best, and * denotes comparable performance to the top result. (paired t-test, p $>$ 0.05)

| Method | $LCEM_2$ | | | BRACS | | |
|---|---|---|---|---|---|---|
| | ACC | AUC | F1 | ACC | AUC | F1 |
| MeanPooling | $69.47_{\pm1.36}$ | $73.70_{\pm4.46}$ | $61.34_{\pm3.07}$ | $65.26_{\pm1.31}$ | $79.78_{\pm2.23}$* | $46.30_{\pm1.74}$ |
| MaxPooling | $74.56_{\pm4.03}$ | $80.66_{\pm4.15}$* | $71.95_{\pm3.91}$ | $66.37_{\pm3.36}$ | $82.01_{\pm2.21}$* | $51.33_{\pm6.46}$ |
| CLAM | $75.38_{\pm1.99}$* | $82.72_{\pm3.90}$* | $71.84_{\pm2.93}$ | $75.50_{\pm1.27}$* | $88.02_{\pm0.95}$* | $62.58_{\pm5.97}$* |
| ABMIL | $71.72_{\pm3.63}$ | $73.75_{\pm8.88}$ | $65.88_{\pm6.12}$ | $74.22_{\pm2.00}$* | $88.06_{\pm2.17}$* | $59.40_{\pm7.14}$ |
| DSMIL | $70.89_{\pm1.47}$ | $73.46_{\pm2.84}$ | $62.95_{\pm2.81}$ | $70.75_{\pm1.76}$ | $86.53_{\pm1.57}$* | $55.88_{\pm3.83}$ |
| TransMIL | $70.65_{\pm15.38}$ | $69.81_{\pm19.32}$ | $55.41_{\pm26.10}$ | $69.30_{\pm5.42}$ | $82.87_{\pm3.83}$ | $52.74_{\pm5.21}$ |
| PMIL | $75.03_{\pm2.18}$* | $82.92_{\pm2.84}$* | $73.45_{\pm3.01}$* | $72.03_{\pm1.81}$* | $87.24_{\pm1.72}$* | $58.74_{\pm7.13}$ |
| CDMIL | **$80.77_{\pm1.89}$** | **$85.02_{\pm0.84}$** | **$76.45_{\pm3.92}$** | $76.83_{\pm1.77}$ | $92.11_{\pm1.65}$ | $65.79_{\pm4.03}$ |

rics. Notably, on the more challenging multi-class BRACS dataset, CDMIL achieves the highest performance in terms of ACC and AUC, demonstrating its strong robustness and versatility across different tasks. These supplementary results further corroborate the findings from our main experiments, underscoring the significant advantages conferred by the integration of causal learning into the vision-language MIL framework.

## C.3 SYMMETRIC GENERALIZATION PERFORMANCE EVALUATION

To further validate the robustness of our CDMIL framework and demonstrate that its strong performance is not specific to a single training domain, we conducted a symmetric generalization experiment. In this setup, we reversed the roles of the $LCEM_1$ and $LCEM_2$ datasets compared to the experiment presented in the main paper. Specifically, all models were trained exclusively on the $LCEM_2$ dataset and then evaluated in a zero-shot manner on the external $LCEM_1$ and TCGA-LUAD test sets.

The results of this symmetric evaluation are detailed in Table 5. Consistent with our primary findings, CDMIL once again substantially outperforms all baseline methods on both external test sets. It achieves a state-of-the-art AUC of 82.50% on $LCEM_1$ and 85.05% on TCGA-LUAD, margins that are significantly ahead of the next-best competitors. This bidirectional validation provides compelling evidence that our causal learning approach enables the model to learn truly domain-agnostic representations, effectively mitigating overfitting to the spurious correlations present in any single training dataset. These findings reinforce the conclusion that the superiority of CDMIL is robust and not merely an artifact of the training data source.

Table 5: **Zero-shot** generalization performance on the EGFR mutation prediction task, with all models trained on the **$LCEM_2$** dataset and evaluated on the external **$LCEM_1$** and **TCGA-LUAD** test sets. (**Notably, due to a severe class imbalance of EGFR mutation types in the TCGA-LUAD dataset, AUC and F1 is considered the primary metrics for a fair comparison on this dataset.**)

| Method | $LCEM_1$ | | | TCGA-LUAD | | |
|---|---|---|---|---|---|---|
| | ACC | AUC | F1 | ACC | AUC | F1 |
| MeanPooling | 45.69 | 58.80 | 44.05 | 60.23 | 63.76 | 50.91 |
| MaxPooling | 40.80 | 60.87 | 31.59 | 76.52 | 69.17 | 58.11 |
| CLAM | 43.89 | 53.56 | 39.51 | 62.12 | 65.27 | 52.26 |
| ABMIL | 53.41 | 57.98 | 53.38 | 56.63 | 67.46 | 48.75 |
| DSMIL | 55.98 | 61.66 | 55.86 | 54.92 | 64.34 | 47.37 |
| TransMIL | 60.75 | 51.72 | 37.79 | 14.02 | 61.23 | 12.29 |
| PMIL | 62.29 | 70.34 | **56.08** | 74.43 | 67.90 | 56.13 |
| CDMIL w/o $\mathcal{L}_{causal}$ | 60.10 | 65.38 | 52.35 | 75.38 | 67.81 | 49.96 |
| CDMIL | **64.35** | **82.50** | 53.51 | **88.26** | **85.05** | **67.16** |

## C.4    Further Visualization

To further demonstrate the robustness and superior interpretability of our CDMIL framework, we provide additional qualitative results on several representative WSIs from the test sets, as shown in Fig. 6. Each row presents a different case, covering a variety of challenging histopathological patterns. The first column displays the original WSI with pathologist-annotated ground-truth contours (in green), followed by the attention heatmaps generated by baseline methods (CLAM, PMIL), our ablation model (CDMIL w/o $\mathcal{L}_{causal}$), and our full CDMIL model.

Across all diverse cases, a consistent pattern clearly emerges: baseline methods such as CLAM and PMIL often produce diffuse or misplaced attention maps, while the CDMIL variant without causal learning is prone to being distracted by spurious features, leading to imprecise attention. In stark contrast, our full CDMIL model consistently generates attention heatmaps that are both focused and accurate. Its high-attention regions are precisely concentrated within the ground-truth contours, exhibiting high concordance with the pathologist's annotations. This advantage holds true even when processing cases with complex tumor boundaries, significant intra-tumor heterogeneity, or large necrotic areas. These visualizations provide compelling visual evidence that our counterfactual causal learning mechanism plays a critical role in guiding the model to focus on true, causally relevant pathological features, thereby enhancing its reliability and trustworthiness for clinical application.

## D    Data Availability

The datasets used in this study are sourced as follows:

- TCGA-LUAD: Slides for the TCGA-LUAD dataset from The Cancer Genome Atlas (TCGA) program, comprising both wild-type and mutant tissues, are publicly available for download from the GDC Data Portal: https://portal.gdc.cancer.gov/.
- Camelyon16: This public challenge dataset is available from its official website: https://camelyon16.grand-challenge.org/.
- BRACS: This dataset can be publicly accessed from its official website: https://www.bracs.icar.cnr.it/.
- Other Datasets: The remaining datasets generated or analyzed during this study are available from the corresponding author upon reasonable request.

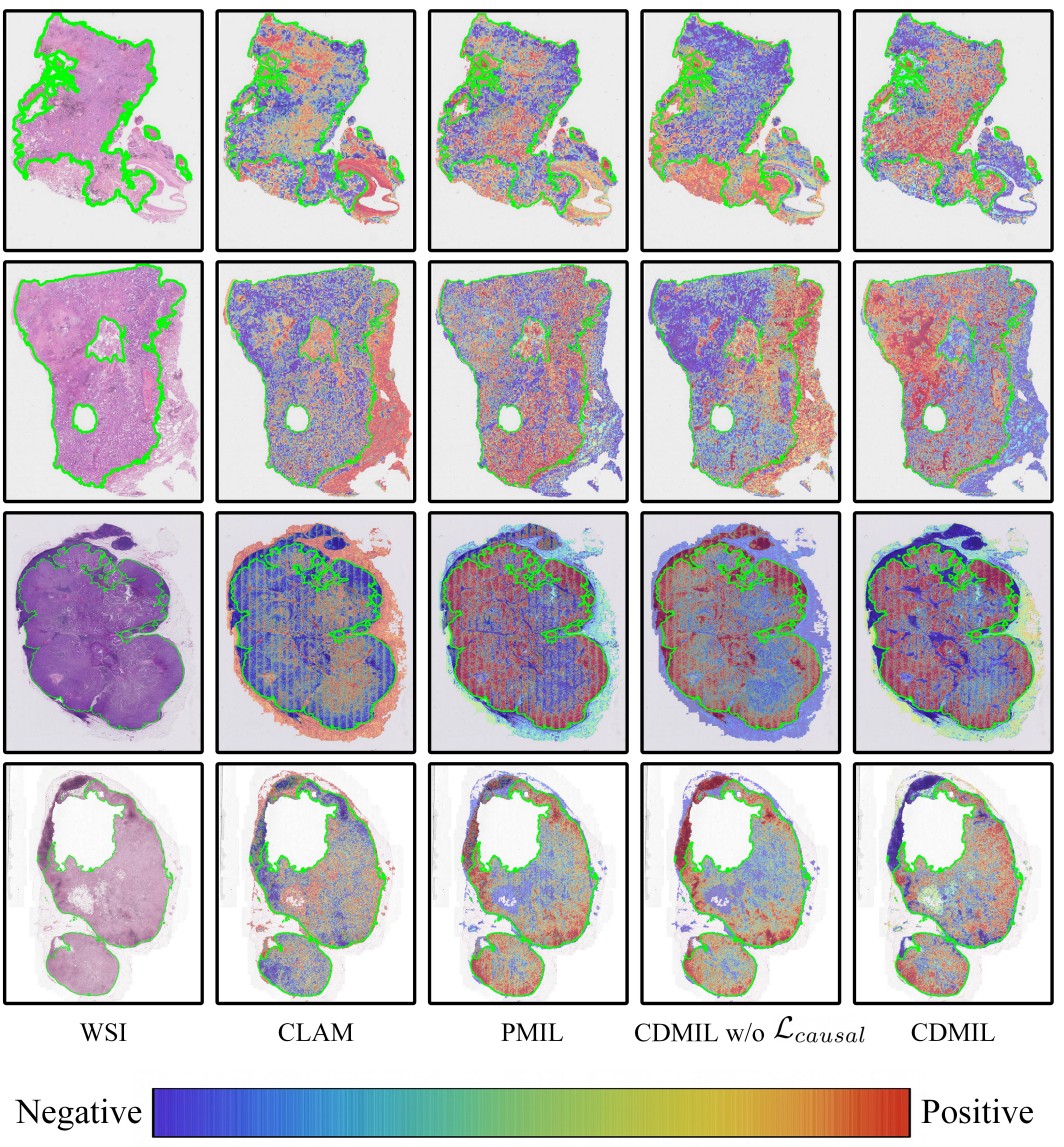

Negative ▬▬▬▬▬ Positive

Figure 6: Qualitative comparison of attention heatmaps, showcasing the superior localization and interpretability of our CDMIL model.

