# OpenReview forum: "Causal Counterfactual Learning in Vision-Language for WSI Classification"
_ICLR.cc/2026/Conference — ICLR 2026 Conference Withdrawn Submission_

### Official Review · Reviewer_xHLs · 2025-10-26

**Soundness:** 2
**Presentation:** 1
**Contribution:** 2
**Rating:** 2
**Confidence:** 2

**Summary:**

The submission proposes a method for improved vision-language alignment, with the motivation of improving out-of-distribution generalization.The core idea seems to involve pushing away the predictive distribution corresponding to a counterfactual combination of visual and text features.

Experiments are conducted on two internal datasets, and two public benchmarks -- TCGA whole-slide LUAD for EGFR prediction, and Camelyon16 for lymph node tumors. The method seems to improve over baseline methods.

**Strengths:**

The question of spurious correlations in vision-language modeling in histopathology is an important one, and ventures towrads improving the alignment is a worthwhile topic.

**Weaknesses:**

There are several points that were quite unclear to me from my reading, and I'm happy to revise my opinion once I understand them.

1. The submission seems to repeatedly suggest that the motivation is that people curate vision-text training data in histopathology by getting LLMs to output diagnostic text, even citing papers such as PathCLIP and PLIP to support this claim. This is typically not done, to my knowledge; contrastive vision-language alignment is performed based on actual diagnostic text in reports or metadata associated with patches or at a whole slide level (as in PathAlign [1]).

2. I do not understand why the method being proposed would be advantageous to a straightforward contrastive learning method like CLIP, where the contrastive loss automatically pushes away misaligned image-text pairs. In fact, CLIP does this for all labels, while the proposed method seems to pick one particular label to push away from the correct pairing, that too, specifically the next one in the list, per Eq. 7.

3. It looks like the proposed method needs a textual input; what happens at test time? Does it see a textual version of the ground-truth, or an LLM-generated version of it? Do the other baseline methods see anything other than the image? (Overall, the description of the method was difficult for me to follow.) Also, if I compare the PMIL numbers reported for Camelyon to that in the paper [2], I see much higher numbers.

[1] PathAlign, MICCAI COMPAYL, https://proceedings.mlr.press/v254/ahmed24a.html

[2] Diagnose Like a Pathologist: Transformer-Enabled Hierarchical Attention-Guided Multiple Instance Learning for Whole Slide Image Classification, IJCAI, https://www.ijcai.org/proceedings/2023/0176.pdf

**Questions:**

Please see above

---

### Official Review · Reviewer_y7FT · 2025-10-27

**Soundness:** 2
**Presentation:** 2
**Contribution:** 2
**Rating:** 2
**Confidence:** 5

**Summary:**

This paper proposes the CDMIL framework, which integrates prototype-guided dual-attention vision–language alignment with a counterfactual causal learning strategy. The paper lacks sufficient novelty, methodological soundness, and empirical rigor.

**Strengths:**

The paper addresses an important topic, causal inference in weakly supervised WSI classification, which is indeed valuable for achieving more interpretable and robust models in computational pathology.

**Weaknesses:**

1. The work lacks clear novelty. Causal inference for WSI classification has already been systematically explored in prior research. However, the related work section omits key causal MIL frameworks such as IBMIL and CaMIL, which are fundamental in this field. Even if the authors claim a “counterfactual causal” perspective, the paper does not provide sufficient grounding or connection to related causal or counterfactual studies.
2. Beyond reporting improved performance, the authors fail to provide a conceptual or theoretical explanation of why counterfactual intervention works in this context. A deeper causal interpretation or ablation demonstrating the mechanism is missing.
3. The comparative experiments rely on relatively early works, whereas several strong WSI methods from the past two years are omitted.
4. The method uses ResNet as the backbone feature extractor, despite mentioning PLIP and other pathology-specific pretrained models in the related work. Given the availability of stronger pretrained encoders (e.g., COACH, UNI), the choice of a generic ResNet seems outdated and suboptimal. Using pathology-aware pretrained encoders would be more appropriate for fair comparison and better feature representation.
5. In Table 2, the F1 score for the binary classification task is only 0.5, which is extremely low and far below results reported in other works.
6. The text ot text feature in the experiment is unclear. Do you send ultra-high-resolution WSIs into GPT?  What text encoder did you used or how did you achieve visual–language alignment. This part of the methodology is confusing and needs clarification.
7. The sentence “visual features of fibrotic stroma act as a confounder” is confusing (Lines 093–095). The authors should clearly define what causal relationship is being modeled and what exactly constitutes the confounder in this setting.
8. Section 3 provides general causal inference background without linking it specifically to the WSI classification task. These are standard causal concepts and do not directly contribute to understanding the proposed method. Moving this section to the appendix would be more appropriate.

**Questions:**

See the weakness.

---

### Official Review · Reviewer_iWS1 · 2025-10-29

**Soundness:** 2
**Presentation:** 2
**Contribution:** 2
**Rating:** 2
**Confidence:** 5

**Summary:**

This paper proposes a causal learning-based scheme to alleviate the spurious textual information introduced by LLMs for further enhancing the performance of WSI classification. To accomplish this, a CDMIL framework is presented. It builds a factual branch and a counterfactual branch and adopts a causal loss to encourage the disagreement of the two branches. In each branch, a visual network and a textual network are used to align WSI and textual features, followed by producing the prediction according to V-L similarity. Three datasets are employed to verify the effectiveness of CDMIL. The empirical results, including the comparison to baselines, features visualization, and ablation studies, show the promising performance of CDMIL in WSI classification.

**Strengths:**

- This paper investigates how to alleviate the spurious information in LLM-generated visual descriptions. It proposes a causal learning-based scheme to address this issue, which seems interesting.
- The experimental results on three datasets show the promising performance of the proposed CDMIL framework. With the proposed causal loss, CDMIL often shows large improvements in WSI classification.

**Weaknesses:**

However, I still have several major concerns, including the possible issues in the framework design and experimental protocol.
- The rationality of the proposed CDMIL framework (important). CDMIL trains new visual and textual networks for robust V-L alignment on very limited WSI samples (often ~1,000). This raises concerns about the goodness of V-L alignment because most VLMs are trained on very large-scale V-L paired samples. In addition, do the authors consider designing the scheme based on existing pathology VLMs (like PLIP or CONCH)? These foundational VLMs exhibit excellent capabilities in V-L alignment. The authors are encouraged to rethink the connections between their work and existing foundational VLMs.
- The experimental setup is inconsistent with commonly used protocols in computational pathology. First, pathology-oriented fundation models (such as UNI, MUSK, and CHIEF) are often adopted as the patch feature extractor for WSIs, instead of the ResNet-50 pretrained on ImageNet. Second, it is recommended to adopt k-fold cross-validation in experiments to ensure a robust evaluation of the algorithms, as most WSI datasets have limited samples.
- Many representative works on WSI classification (such as DTFD-MIL, R2T-MIL, and so on) are missing in the compared baselines. In Table 1, only PMIL was proposed in 2025 and the remaining ones are before 2022. The authors are strongly encouraged to add representative baselines for comparisons to make the experimental results sound enough.
- How is the impact of $\lambda$ on the performance of CDMIL?

**Questions:**

Please see Weaknesses

---

### Official Review · Reviewer_GVdn · 2025-10-30

**Soundness:** 2
**Presentation:** 3
**Contribution:** 2
**Rating:** 4
**Confidence:** 3

**Summary:**

This paper addresses the introduction of spurious correlations between visual signals and textual descriptions by proposing a Causal-learning Dual-attention MIL (CDMIL). This framework first achieves initial visual-text alignment through prototype-guided dual attention, then employs a counterfactual learning strategy for causal intervention. During this process, counterfactual text replaces the original text, enabling the model to overcome its dependence on spurious correlations and learn genuine causal relationships. Experimental results demonstrate that CDMIL achieves state-of-the-art performance in both classification accuracy and out-of-distribution robustness, validating the effectiveness and superiority of this causal learning framework.

**Strengths:**

Originality. This paper describes WSI-level pathological diagnosis as a three-layer model of "visual-linguistic-causal": on the one hand, through a prototype-driven dual attention mechanism, the pathological semantics generated by LLM are explicitly aligned to local tile representations; on the other hand, through counterfactual text substitution and causal regularization, the model actively punishes text descriptions that rely on spurious correlations. This approach to causal inference is a novel perspective, especially in the high-risk field of computational pathology, where it is rarely systematically described.

Quality. CDMIL first uses prototype-based cross-modal alignment, then introduces counterfactual branches and performs causal regularization using KL discriminant, finally completing classification at the slice level. The experimental results not only achieve state-of-the-art performance on commonly used WSI benchmarks but also demonstrate effectiveness in cross-distribution generalization. Ablation studies also show that relying solely on text can improve the mean but is unstable, while adding causal regularization improves both performance and stability simultaneously.

Clarity. The paper is written in a relatively clear manner, with necessary calculation formulas in each section, making the modules easy to understand. The descriptions of experimental details such as training data and hyperparameters are sufficiently transparent, and reproducibility appears feasible.

Significance. Traditional WSI MIL methods often rely solely on visual patch features for classification. While recent cross-modal methods have incorporated LLM text interpretability, they can still be affected by the illusions created by LLM. CDMIL doesn't simply pursue higher AUC; it's designed for robust generalization and causal consistency. In particular, it significantly improves OOD results, demonstrating a reduction in overfitting and spurious associations. This is particularly significant for a domain where deployment credibility is a core requirement.

**Weaknesses:**

1. The prototype-driven dual-attention visual-language alignment is essentially a combination of prototype-based queries and two-stage cross-modal attention aggregation, highly similar in concept to existing prototype-based MIL, cross-attention fusion, and visual-text alignment pipelines. The counterfactual branch replaces the original text with average text features from another category and adds distribution difference regularization, similar to the existing training idea of ​​applying adversarial or counterfactual perturbations to the input modality and constraining prediction sensitivity. The paper lacks clear evidence demonstrating which parts represent new mechanisms that previous methods couldn't achieve, and which merely package known elements, such as prototype-based queries, cross-modal attention, and adversarial intervention regularization, into a pathological diagnosis scenario. Therefore, overall, it seems more like a domain adaptation of a mature approach than a proposal of a new learning paradigm.

2. The ablation analysis only demonstrates high-level comparisons such as performance degradation after removing causal regularization and only visual-text alignment has larger variance, without systematically performing component-level ablation, such as removing the visual prototype stage, removing the text prototype stage, removing any stage of the two-stage attention, or retaining only counterfactual regularization.  Therefore, it cannot determine which module is the key driver of OOD improvement.

3. The paper claims that "our method focuses on real tumor evidence rather than pseudo-features such as fibrosis," but it does not provide quantitative error analysis or region-level validation. For example, it does not specifically demonstrate the frequency and visualization results of the baseline misclassifying fibrosis as tumor while CDMIL does not misclassify it. The model's ability to learn causal pathological features remains inferential, without direct experimental support.

4. The paper emphasizes deployability and robustness, but does not provide resource assessments such as training and inference costs, memory usage, and dual-branch overhead. It also does not verify whether these hyperparameters can be transferred to different backbones or multiple scenarios without extensive retuning.

5. While the paper's framework is clearly decomposed, including visual prototype query, text prototype modulation, counterfactual text library, and KL regularization, key implementation details are not clearly described and seem to lack quantitative explanation in the current manuscript. For example: how is the text feature library constructed and frozen? Is the class mean updated or estimated in a single round?     How is the counterfactual class selected in multi-class tasks? Currently, it is described as (Y+1) mod C, which is very coarse. Is the coefficient λ of the KL difference term sensitive? Does it need to be retuned for different datasets?

6. This paper refers to text replacement as counterfactual intervention and KL difference as causal regularization. This aligns with the intervention intuition in causal inference, but it still appears to be empirically heuristic, failing to demonstrate that maximizing the difference between fact and counterfactual necessarily suppresses confounding rather than creating new biases. Currently, the paper's description of causality is difficult to classify as rigorous causal identification.

**Questions:**

1. In the dual-attention module, are both stages of interaction, visual prototype and textual prototype, essential?   Please provide further component-level ablation, removing the visual prototype stage and the textual modulation stage separately, to quantify their respective impact on main task performance and OOD performance.

2. During the inference stage, is it necessary to run both factual and counterfactual branches simultaneously, or is the counterfactual branch only used for training?   What is the exact inference cost?

3. Is counterfactual intervention still reasonable in multi-class settings?   When the categories are not binary, does a simple "(Y+1) mod C" replacement still construct meaningful contradictory pairs?   Have you considered selecting the most confusing counterfactual category instead of a fixed rotation?

---

### Note · Authors · 2025-12-01

I have read and agree with the venue's withdrawal policy on behalf of myself and my co-authors.